# Implant Treatment by Guided Surgery Supporting Overdentures in Edentulous Mandible Patients

**DOI:** 10.3390/ijerph182211836

**Published:** 2021-11-11

**Authors:** Eugenio Velasco-Ortega, Alvaro Jiménez-Guerra, Ivan Ortiz-Garcia, Nuno Matos Garrido, Jesús Moreno-Muñoz, Enrique Núñez-Márquez, José Luis Rondón-Romero, Daniel Cabanillas-Balsera, José López-López, Loreto Monsalve-Guil

**Affiliations:** 1Department of Stomatology, Faculty of Dentistry, University of Seville, C/Avicena s/n, 421009 Seville, Spain; evelasco@us.es (E.V.-O.); alopajanosas@hotmail.com (A.J.-G.); ivanortizgarcia1000@hotmail.com (I.O.-G.); nunogarrido@orallagos.pt (N.M.G.); je5us@hotmail.com (J.M.-M.); enrique_aracena@hotmail.com (E.N.-M.); jolurr001@hotmail.com (J.L.R.-R.); danielcaba@gmail.com (D.C.-B.); lomonsalve@hotmail.es (L.M.-G.); 2Department of Odontostomatoly, Faculty of Medicine and Health Sciences (Dentistry), University of Barcelona, C/ Feixa LLarga s/n, 08907 Hospitalet de LLobregat, Spain

**Keywords:** guided implant surgery, overdenture, early loading, edentulous mandible

## Abstract

Introduction: This study aimed to show the clinical outcomes of implants inserted by guided surgery supporting mandibular overdentures in edentulous patients. Patients and methods: Mandibular edentulous patients were diagnosed with an oral examination, cone-beam computerized tomography, and diagnostic casts for intermaxillary relations and treated with overdentures over two implants by guided surgery. After flapless surgery, implants were early loaded with an overdenture at 6 weeks. Results and discussion: Fourteen patients (nine females and five males) were treated with 28 implants. Four patients (28.6%) had a previous history of periodontitis. Five patients (35.7%) were smokers. Nine patients (64.3%) suffered from systemic diseases (i.e., diabetes, cardiovascular diseases). The clinical follow-up of the study was 44.7 ± 31.4 months. Clinical outcomes showed a global success of 100% of implants. Fourteen overdentures were placed in the patients over the implants. Mean marginal bone loss was 1.25 mm ± 0.95 mm. Four patients (28.6%) showed some kind of mechanical prosthodontic complications. Six implants (21.4%) were associated with peri-implantitis. Conclusions: This study indicates that treatment of mandibular edentulous patients with overdentures by guided surgery and early loading of implants placed appears to be a successful implant protocol.

## 1. Introduction

The prosthodontic rehabilitation of edentulous patients by implant-supported restorations is considered a predictable and successful modality of dental treatment [1,2]. The implant-supported overdenture has become the therapy of choice for the edentulous mandible in geriatric patients [3]. Implant-supported overdentures have been proven to be an effective treatment alternative for restoring fully edentulous patients with a high success rate [4]. Moreover, implant-supported mandibular overdentures provide patients with a significant improvement of oral quality of life compared with conventional removable dentures [5].

Guided implant dentistry has rapidly grown in popularity, and is now widely used as a real alternative to planning for surgical and prosthodontic steps of implant treatment [6,7]. During the last decades, the incorporation of three-dimensional imaging technology by the introduction of cone-beam computed tomography (CBCT) allowed the acquisition of the bone volume and density of the jaws in a simple protocol and with a considerable reduction of radiation absorbed by the patient [8,9].

The development of specific software in clinical guided implant dentistry provides a virtual method to treatment planning of flapless implant surgery in rehabilitation of totally and partially edentulous patients [10,11]. The accuracy of positioning of the implants in alveolar ridges must be considered an important factor in clinical situations with limited bone volume. In geriatric edentulous patients, the loss of mandibular teeth induces many anatomical changes, particularly regarding its shape, volume, and density. The diagnosis of these bone characteristics by CBCT allows a clinician to select the best implant surgery to achieve good primary stability and fast osseointegration [12,13]. 

Several studies have reported excellent clinical outcomes of guided implant treatment in fully edentulous patients, demonstrating that this comprehensive approach has a good implant survival and that implant-supported fixed prosthesis can be delivered to the patient [14,15]. The use of CBCT to plan the insertion of implants by surgical guides, aided by specific software, includes the use of a flapless technique with an important reduction of surgery times, postoperative pain, and better patient comfort [16,17]. 

However, only a few clinical studies have evaluated the efficacy of guided implant dentistry in the treatment of edentulous patients with overdentures [18,19]. There are only several case reports available to document the feasibility of this technique for overdentures [20,21,22,23,24]. This technique of guided flapless implant surgery has been suggested for the geriatric edentulous mandibular patients because CBCT allows that certain anatomical structures (mental foramina, inferior alveolar nerve) are easily identified and protected [12,13,18,21]. The prosthetic approach with overdenture induces the choice of implant position whose characteristics (shape, surface, thread) are consistent with the underlying bone diagnosis (volume and cortical density and trabecular spaces) [20,22]. Additionally, guided implant dentistry minimizes the possibility of postoperative soft tissue loss and reduces the surgery duration compared to patients with conventional flaps [23]. 

This study aimed to investigate the clinical results of guided surgery of implants and early loading with mandibular overdentures in the treatment of edentulous patients. 

## 2. Materials and Methods

This study was carried out at the Faculty of Dentistry of the University of Seville during the years 2013 to 2019. Due to the nature of clinical research, the principles described in the Declaration of Helsinki were taken into account for the design of the study. The approval of the ethics committee of the University of Seville and the informed consent of the patients were also obtained.

The inclusion criteria were edentulous lower jaw patients in need of rehabilitation with mandibular overdentures implant-supported. The study population consisted of 9 women and 5 men, aged between 53 and 85 years, resulting in a total of 14 patients with a mean age of 70.6. The exclusion criteria were the following: (a) presence of chronic systemic disease, such as uncontrolled diabetes mellitus or coagulation disorders; (b) harmful habits such as smoking with consumption greater than 10 cigarettes/day, alcoholism, or drug use; (c) oral conditions such as uncontrolled periodontal disease and bruxism; previous history of periodontitis but in a situation of controlled disease are not excluded. Regarding treatment planning, additionally to the intraoral examination and clinical photographs, CBCT and diagnostic models for intermaxillary relationships were performed. Possible implant treatment options were explained to the patients, choosing the implant-supported overdenture through guided surgery.

The patients were prescribed an antibiotic treatment consisting of 500 mg of amoxicillin and 125 mg of clavulanic acid 1 h before the intervention, as well as every 8 h/7 days after the treatment. As analgesic regimen, ibuprofen 600 mg/6 h/7 days was indicated. For the following 30 days, a chlorhexidine mouthwash was prescribed, 2 times per day. Articaine with adrenaline as a vasoconstrictor was injected as local anesthesia for implant treatment.

All participants underwent cone-beam computer tomography (Picasso Master 3D^®^, Vatech, Gyeonggi-do, Korea) with a scan prosthesis and occlusal index positioned in the mouth. The implants were planned in 3D software (Galimplant 3D ^®^, Galimplant, Sarria, Spain) in the optimal position considering both the alveolar process and the prosthetic demands (Figure 1 and Figure 2). 

A flapless surgical approach was chosen with the help of a guided surgical template. The design of this surgical template was performed using digital planning (Figure 3). In all patients, the template was secured to the underlying bone with two screws in the vestibular plates to avoid movement during the surgery. Guided surgery began with the preparation of two surgical beds, through the sequential steps of drills according to a protocol of progressive diameter increase. Finally, the implants were inserted positioned by the guide.

The implants selected for insertion in the patients were Surgimplant ^®^ (Galimplant, Sarria, Spain), which were characterized by an internal connection and a surface treatment consisting of sandblasting and acid etching. Insertion torque and resonance frequency were analyzed to determine implant stability after placement. Insertion torque was measured before the removal of the surgical guide. An insertion torque ≥35 Ncm was considered adequate for implant stability at the time of placement. [8,14]. Finally, resonance frequency analysis was used to confirm the stability of each implant immediately after removal of the guide once the implants have been placed. Adequate primary stability required an ISQ (Implant Stability Quotient) value between 55 and 85 [15]. However, an ISQ value greater than 65 was considered to perform a one-stage surgery and an early loading protocol (Table 1) [25]. After the surgical procedure, healing abutments were connected to the implants. Sutures were not used in any patient. 

Existing dentures were molded and relined with soft material to not interfere with the peri-implant tissues and reduce occlusal forces on the implants. After six weeks of surgery, early loading was performed. An open impression technique with individualized tray and addition silicone material was used. An Overdent ^®^ (Galimplant, Sarria, Spain) attachment system was used in the manufactured and retention of the overdentures over the osseointegrated implants. (Figure 4 and Figure 5).

Implant stability and absence of peri-implant radiolucency, mucosal suppuration, and pain were considered as survival criteria. All patients were included in a maintenance program consisting of clinical and radiological examination and cleaning of prostheses and implants. The frequency of revisions was set at 3 and 6 months and annually after guided implant insertion. Periapical radiographs acquired digitally using positioners were used to measure follow-up in marginal bone loss. The analyzed variables included patient information (gender, age, dental health, systemic diseases, history of periodontitis, smoking habit), details about the placed implants (type, number, position, diameter, and length), and the implant-supported overdenture including the dates of delivery. In addition, surgical, biological, and technical complications that occurred during implant insertion, postoperatively, or during function in the follow-up period, were recorded.

A statistical analysis of the variables obtained was carried out using SPSS software (SPSS 11.5.0, SPSS, Chicago, IL, USA). Descriptive statistics were used to report the clinical results of the study. Absolute and relative percentage frequencies of qualitative variables were obtained, and chi-square test was used to analyze distributions. Means, standard deviations (SD), medians, ranges, and 95% confidence intervals (CI) were obtained for the quantitative variables. An analysis of variance (ANOVA) was used to confirm the similarities in the groups. The analysis of differences between the groups created based on the different risk factors measured was performed using the non-parametric Mann–Whitney U test. The level of statistical significance was established for a value of *p* < 0.05.

## 3. Results

Twenty-eight implants were inserted in 14 edentulous mandible patients, 9 females, and 5 males. No significant statistical differences were found related to sex and age (chi-square test, *p* = 0.87208). Four patients (28.6%) had a previous history of periodontitis. Five patients (35.7%) were smokers (Table 2). Nine patients (64.3%) exhibited medical conditions (i.e., diabetes, cardiovascular diseases).

The average follow-up period was 44.7 ± 31.4 months (ranged: 12–84 months). Twenty-eight implants (100%) had a diameter of 4 mm. Twenty implants (71.4%) were 10 mm in length, and 8 (28.6%) were 12 mm. No implant was lost during the follow-up (Table 3). The cumulative survival rate for all implants was 100%. 

During the follow-up period, four implants (14.3%) in two patients (14.3%) were associated with peri-implantitis (Table 4). Peri-implantitis was more frequent in those patients with a previous history of periodontitis (50%) and smoking patients (40%). 

The mean marginal bone loss was 1.25 ± 0.94 mm, ranging from 0.8 to 1.7 mm during the follow-up evaluation (Table 5). In patients less than 70 years, the marginal bone loss was 0.91 ± 0.88 mm compared with 1.50 ± 0.80 for more than 70 years, with statistical differences (ANOVA; *p* = 0.0077) (Table 5). 

In female patients, the marginal bone loss was 1.21 ± 0.90 compared with 1.32 ± 1.04 in male patients, without statistical differences (ANOVA; *p* = 0.6763) (Table 5).

In patients with a history of periodontitis, the marginal bone loss was 1.40 ± 1.10 compared with 1.19 ± 0.88 in patients without a history of periodontitis, without statistical differences (ANOVA; *p* = 0.4436) (Table 5).

In smoking patients, the marginal bone loss was 1.32 ± 1.04; while that in patients without smoking habits was 1.21 ± 0.90 mm. No significant statistical differences were found (ANOVA; *p* = 0.6763) (Table 5).

In patients with systemic diseases, the marginal bone loss was 1.38 ± 0.93; and 1.00 ± 0.86 for patients without medical conditions without statistical differences (ANOVA; *p* = 0.1173) (Table 5).

In patients with a follow-up less than 5 years, the marginal bone loss was 0.93 ± 0.81 compared with 1.66 ± 0.58 with a follow-up more than 5 years, with statistical differences (ANOVA; *p* = 0.0001) (Table 5).

After 6 weeks of healing period, 14 overdentures were performed over 28 implants placed in the patients. Mechanical prosthodontic complications were recorded in four patients (28.5%). (Table 4). Two patients (14.2%) showed resin fracture of a prosthesis, and two patients (14.2%) needed the change of locator-attachment system.

## 4. Discussion

The aim of this study was to evaluate clinical outcomes in planning and treatment by implant-guided surgery of edentulous patients with a mandibular overdenture. The rehabilitation of edentulous mandible patients constitutes an important challenge because optimal implant planning is strongly related to a correct diagnosis to achieve a successful prosthodontic rehabilitation [14,15]. 

The imaging diagnosis by CBCT allows the clinician to reduce the risk of damaging vital structures in mandibular areas with limited residual bone. The surgical placement of implants in the anterior edentulous mandible for overdenture treatment is a predictable option with long-term successful results. However, implant-guided surgery in the atrophic anterior mandible presents several anatomic challenges owing to the vascular and neurologic structures related to this region. The incorporation and development of CBCT have resulted in a better appreciation of the risks involved with surgery in this area [14,15]. 

Moreover, in the anatomical diagnosis of the edentulous mandible, the assessment of the bone volume and density of the mandible by CBCT allows a practitioner to select the best implant design and surface to achieve good primary stability and speed osseointegration. All patients of the present study were evaluated, by CBCT, before the surgery to assess the specific bone characteristics, as cortical density and trabecular spaces, choosing the best location for the placement of implants, also according to prosthodontic approach with overdentures [12].

In the present study, based on the clinical and CBCT findings and the patient’s expectations, treatment planning was established to place two implants in the anterior mandible by using flapless surgery. Fourteen patients received 28 implants, inserted through a flapless guided surgery. No implants were lost, during the mean clinical follow-up of 44.7 ± 31.4 months. The cumulative implant survival rate was 100%. Flapless implant surgery is a minimally invasive surgical approach that has several important advantages for both the clinician and the patient [14,15]. Flapless guided implant surgery increases the ability to insert implants more precisely, specifically in fully edentulous cases, with an important reduction of surgery duration, better clinical conditions after surgery, and the possibility of placing restorations for immediate loading. Computer-guided dental implant systems also provide necessary information for the prosthetic evaluation of mandibular edentulous patients [18,19,20,21,22,23,24].

Evidence provided by implant-supported mandibular overdenture research on different loading protocols is important. Several systematic reviews have been published about the clinical applicability of conventional-, early-, and immediate-loaded implants for overdenture treatment. Among the studies, the mandibular overdenture design included the use of two, three, and four implants and different attachment systems as locator, bar, and balls [26,27,28]. 

The mean MBL found by us (1.25 mm) is higher than that reported by Galindo et al. in 2015 [29], who published a 0.6 mm loss (mean distal and mesial) at 6 months; and 1.11 mm at 18 months (mean distal and mesial). Our study has a mean follow-up of 44.7 months (±31.4), and if we analyze other studies, such as the one by Palacios-Garzón et al. [30], they accept a loss of less than 2 mm in the first year. However, we must not forget that losses greater than 0.5 mm in the first year increase the risk of presenting peri-implantitis in the long term by 5.43 times [31]. Both in the cases of peri-implantitis, and with regard to the MBL data, we have not been able to relate them to the height of the locator-attachment system used, all of which is that the literature refers to a higher bone order in the lower systems [32,33].

Computer-guided surgery in the rehabilitation with overdentures, by immediate loading of four implants, of edentulous mandibular patients has been investigated in several studies [18,19,24,28]. During a follow-up of 2 years, 10 consecutive patients were restored with implant-supported overdentures [18]. Patients were treated with four intraforaminal implants using a computer-guided flapless approach. No implant was lost. Patients demonstrate the ability of oral hygiene for maintenance of peri-implant tissue health. The satisfaction of patients was very high. These results reported that computer-assisted implant dentistry can be a predictable protocol for treating elderly edentulous patients with a mandibular overdenture [18]. These satisfying results were confirmed by a recent one-year randomized controlled clinical trial [19]. Thirty mandibular edentulous patients were rehabilitated using overdentures supported by four implants, inserted by guided surgery in canine and second premolar position. These patients were randomly distributed into two groups, performing an immediate loading protocol using resilient stud (locator) or stress-free implant bar (SFI-Bar) attachments. After a one-year follow-up, implant survival in the locator group was 96.6%, while in the group using SFI-Bars, it was 98.3%. The locator showed significantly higher overall satisfaction, satisfaction with retention, comfort, and cleanliness of overdentures compared to bar-overdenture-rehabilitated patients. [19].

Immediate loading of two implants in overdentures by guided surgery, in edentulous mandibular patients, has been investigated in some clinical cases [22,23]. In these clinical situations was necessary high primary implant stability measured by resonance frequency analysis. Implant stability quotient values over 65, which was a prerequisite for the immediate loading, should be accomplished. Locator abutments were inserted on the implants and the metal housings were seated on the locators [23]. Another type of used retention system of overdentures was magnetic attachments [22].

In the present study, after guided placement of implants, the treatment planning was the early delivery of implant-retained overdentures at 6 weeks. No implants were lost, during the clinical follow-up. The cumulative implant survival rate was 100%. Early loading of two implants placed in edentulous mandibular patients can be a reliable and predictable technique for implant-supported overdentures [26,27]. Although all three loading protocols (conventional, early, immediate) provide high survival rates of treatment with overdentures, early- and conventional-loading protocols are still better documented than immediate loading and seem to result in fewer implant failures during the first year of clinical follow-up [26].

Marginal bone loss in overdentures by guided surgery implants has been reported in a recent study [19]. The marginal bone loss of immediately loaded implants ranged from 0.68 to 0.83 mm after a 1-year clinical follow-up. Similar mean values were obtained in locator (0.83 ± 0.10 mm) and bar retention systems (0.87 ± 0.13 mm). Marginal bone loss was significantly higher at 12 months compared with 6 months. This increased marginal bone loss can be due to the bone response to overdenture loading and bone maturation combined with functional occlusal forces [19]. In the present study, the mean marginal bone loss was 1.25 ± 0.94 mm during a follow-up period of 44.7 ± 31.4 months. Clinical outcomes showed a higher significant marginal bone loss in patients for more than 70 years and followed more than 5 years. Additionally, a common history of medical conditions (64.3%), smoking (35.7%), and periodontitis background (28.6%) can explain the higher mean values of marginal bone loss in the patients of the present study.

The global satisfaction of patients with computer-guided surgery and prosthetic rehabilitation with overdentures is very high because the postoperative pain and discomfort is very low and improves the compliance in the functional and aesthetic outcomes of prosthodontic treatment [5,18,24]. However, biological and technical complications had also been reported in several studies of treatment with implants placed using computer-guided surgery [28,34]. In the present study, four implants (14.3%) in two patients (14.3%) were associated with peri-implantitis. Peri-implantitis was more frequent in those patients with a previous history of periodontitis (50%) and smoking patients (40%). Additionally, the locators abutments can show an important vertical bone loss with higher plaque and gingival scores and increased probing depth with the progress of time that can explain the development of peri-implantitis in susceptible patients (i.e., periodontitis background and smoking habits) [19]. 

Prosthodontic complications were frequent in the present study. Four patients (28.5%) showed some kind of technical complications. Two patients (14.2%) showed resin fracture of the prosthesis, and two patients (14.2%) needed the change of locator-attachment system. The matrix resiliency among different locator systems can eventually be compromised by insertion and removal of the overdenture. This prosthetic complication may require reactivation or even replacement of the matrix; however, this maintenance procedure can be easily provided by the clinician in clinical practice [27].

## 5. Conclusions

Guided implant dentistry is widely used as a comprehensive alternative to planning for surgical and prosthodontic steps of treatment of edentulous patients. This study indicates that the treatment of edentulous patients with mandibular overdentures by guided surgery and early loading of implants placed appears to be a successful implant protocol. 

## Figures and Tables

**Figure 1 ijerph-18-11836-f001:**
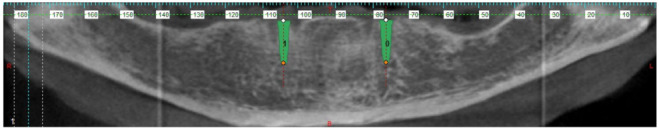
Planning example using the software (Galimplant 3D^®^, Galimplant, Sarria, Spain), which allows for choosing the right position. Orthopanoramic cut.

**Figure 2 ijerph-18-11836-f002:**
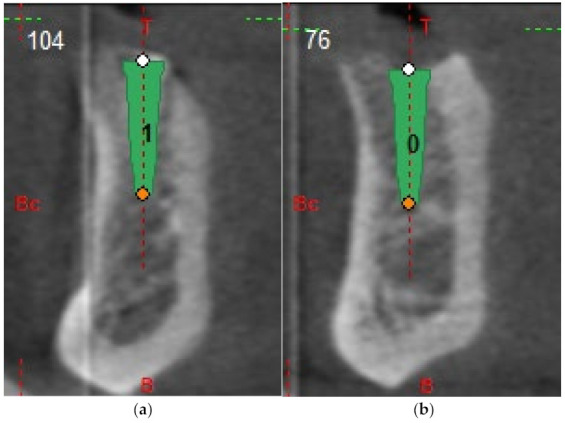
(**a**,**b**) Planning example using the software (Galimplant 3D^®^, Galimplant, Sarria, Spain), which allows for choosing the right position. Axial cuts.

**Figure 3 ijerph-18-11836-f003:**
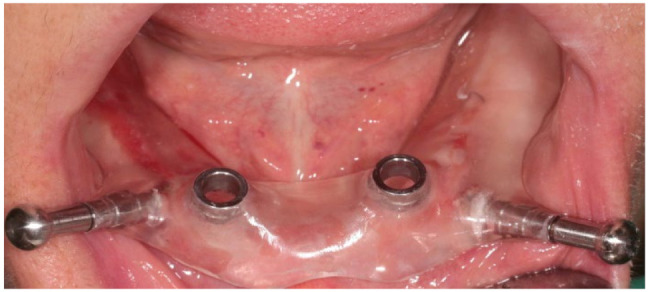
Placement of the surgical splint with the rings that allow the insertion of the implants through a flapless approach.

**Figure 4 ijerph-18-11836-f004:**
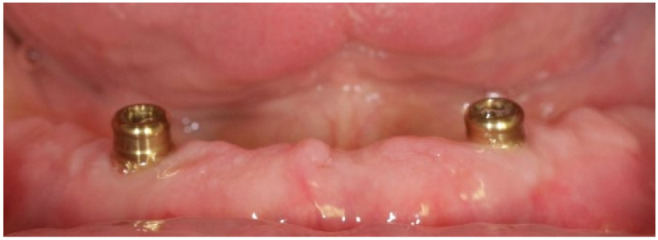
Clinical appearance after placement of prosthetic attachments.

**Figure 5 ijerph-18-11836-f005:**
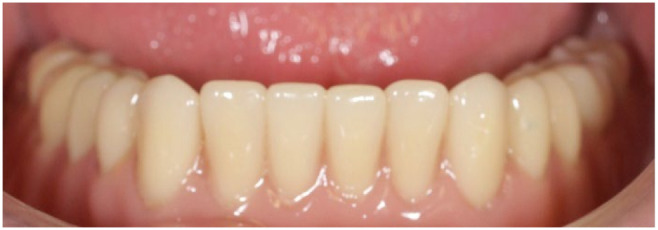
Inserted prosthesis.

**Table 1 ijerph-18-11836-t001:** ISQ (Implant Stability Quotient) value and clinical recommendation.

	ISQ Value	Clinical Recommendation
Primary stability inadequate	<55	Increase implant stability
Low stability	55–60	2-stage surgery/ Conventional loading
Medium stability	60–69	Possible 1-stage surgery/ Early loading
High stability	>70	Immediate loading

**Table 2 ijerph-18-11836-t002:** Description of patients’ characteristics.

	*n*	%
Females	9	64.3
Males	5	35.7
History of periodontitis	4	28.6
Smokers	5	35.7
Medical conditions	9	64.3

*n* = patient.

**Table 3 ijerph-18-11836-t003:** Description of implant’s characteristics.

	*n*	%
4 mm implant diameter	28	100
10 mm implant length	20	71.4
12 mm implant length	8	28.6
Loss of implant	0	0

*n* = implant.

**Table 4 ijerph-18-11836-t004:** Description of patients with complications.

	*n*	%
Implant loss	0	0
Peri-implantitis	2	14.3
Technical complications	4	28.6

*n* = implant.

**Table 5 ijerph-18-11836-t005:** Mean marginal bone loss of patients.

Age *	≤70 years	>70 years	
	0.91 ± 0.88	1.50 ± 0.80	*p* = 0.0077
Gender	Female	Male	
	1.21 ± 0.90	1.32 ± 1.04	*p* = 0.6763
History of periodontitis	+	-	
	1.40 ± 1.10	1.19 ± 0.88	*p* = 0.4436
Smokers	+	-	
	1.32 ± 1.04	1.21 ± 0.90	*p* = 0.6763
Medical conditions	+	1.25 -	
	1.38 ± 0.93	1.00 ± 0.86	*p* = 0.1173
Follow-up *	≤5 years	>5 years	
	0.93 ± 0.81	1.66 ± 0.58	*p* = 0.0001
Total	1.25 ± 0.94 (0.8–1.7)		

(*) Statistically significant.

## Data Availability

The associated data of a statistical nature can be requested to: jl.lopez@ub.edu & evelasco@us.es.

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
