# Peer review of "Implant Treatment by Guided Surgery Supporting Overdentures in Edentulous Mandible Patients"

_ijerph, 2021, doi:10.3390/ijerph182211836_

Round 1

Reviewer 1 Report

First of all I would like to congratulate the team for the excellent clinical protocols and very good research conducted.

I would like to present a few suggestions to improve the quality of the manuscript:

  1. in the abstract(line 23) you state that some patients had previous history of periodontitis and at line 87 you exclude patients with periodontal disease. I would recomand that you further develop the idea to be more specific.
  2.   It would help the readers to better understand the dynamics of the ISQ measurements if they were summarised in a table. There may be a correlation between low insertion torque, ISQ and complications that the investigators may find.
  3. using guided surgery implies that you accurately place dental implants. you do not state any measurments that overlap the planning at the postop imaging.
  4. the guides for implant loading using ISQ which the authors cite ( 15 ) do not accurately present the loading protocol for dental implants using ISQ. Research and recomandations say that ISQ at 55 you need two stage surgery and reevaluation of the ISQ after 2 months before loading. we belive that this is very important for the research and needs improving.

Author Response

# First of all, I would like to congratulate the team for the excellent clinical protocols and very good research conducted.

I would like to present a few suggestions to improve the quality of the manuscript:

1.-in the abstract (line 23) you state that some patients had previous history of periodontitis and at line 87 you exclude patients with periodontal disease. I would recomand that you further develop the idea to be more specific.

Response:

Thank you very much for the comments made. We have tried to improve the work according to your suggestions:

The fundamental aspect of the periodontal patient is its treatment. Therefore, as an exclusion criterion, we indicated uncontrolled periodontal patients, we have added [in red, line 88-90]

., c) oral conditions such as uncontrolled periodontal disease and bruxism; previous history of periodontitis but in a situation of controlled disease are not excluded. Regarding…

In these conditions, 4 patients with a history of sustained periodontitis and on maintenance therapy were included, as shown in the results. [Line 171]: Four patients (28.6%) had a previous history of periodontitis.

2.-It would help the readers to better understand the dynamics of the ISQ measurements if they were summarised in a table. There may be a correlation between low insertion torque, ISQ and complications that the investigators may find.

Response:

  1. A new text, and table and reference, table has been introduced in this aspect: [In red, line 128-134]

….However, an ISQ value greater than 65 is considered to perform a one-stage surgery and an early loading protocol (Table 1) [25]. After the surgical procedure, healing abutments were connected to the implants. Sutures were not used in any patient.

Table 1. ISQ (Implant Stability Quotient) value and clinical recommendation.

ISQ value

Clinical recommendation

Primary stability inadequate

< 55

Increase implant stability

Low stability

55 – 60

2-stage surgery / Conventional loading

Medium stability

60 – 69

Possible 1-stage surgery/

Early loading

High stability

> 70

Immediate loading

The new reference is, [line 390]

  1. Baltayan S, Pi-Anfruns J, Aghaloo T, Moy PK. The Predictive Value of Resonance Frequency Analysis Measurements in the Surgical Placement and Loading of Endosseous Implants. J Oral Maxillofac Surg 2016, 74, 1145-1152. DOI: 10.1016/j.joms.2016.01.048.

3.-using guided surgery implies that you accurately place dental implants. you do not state any measurments that overlap the planning at the postop imaging.

Response:

Thank you very much for your comment. We agree with you that the use of guided surgery improves the precision of implant placement, especially important in fixed rehabilitation and with a large number of implants. However, the specific objective of this study was to investigate the clinical outcomes of guided implant surgery and early loading with mandibular overdentures in the treatment of edentulous patients.

4.-the guides for implant loading using ISQ which the authors cite ( 15 ) do not accurately present the loading protocol for dental implants using ISQ. Research and recomandations say that ISQ at 55 you need two stage surgery and reevaluation of the ISQ after 2 months before loading. we belive that this is very important for the research and needs improving.

Response:

In addition to the new table [Table 1], a detailed description regarding the usefulness and clinical recommendations of the ISQ measurement has been incorporated into the manuscript: [line 128-129]

 “Adequate primary stability requires an ISQ (Implant Stability Quotient) value between 55 and 85 [15]. However, an ISQ value greater than 65 is considered to perform a one-stage surgery and an early loading protocol (Table 1) [25]”

Reviewer 2 Report

In the Discussion section:

The very significant mean MBL (1.25 mm) described in this article should be underlined. In fact, MBL > 0.44mm at 6 months is an indicator of bone loss progression over time (18 months), as demonstrated by Galindo-Moreno

Galindo-Moreno P, León-Cano A, Ortega-Oller I, Monje A, O Valle F, Catena A. Marginal bone loss as success criterion in implant dentistry: beyond 2 mm. Clin Oral Implants Res. 2015; 26(4): e28-e34.

In addition, implants with MBL≥0.5 mm during the first year of function showed a 5.43 times higher odds for future peri-implantitis development and implants in patients with smoking habits and a history of periodontitis are at the highest risk of developing peri-implantitis and experiencing implant loss if early bone loss exceeds the threshold of 0.5 mm at 1 year.

Windael S, Collaert B, De Buyser S, De Bruyn H, Vervaeke S. Early peri-implant bone loss as a predictor for peri-implantitis: A 10-year prospective cohort study. Clin Implant Dent Relat Res. 2021; 23(3): 298-308.

The role of the Locator abutment height on MBL should be discussed and (eventually) investigated, based on these recent articles: 1- Spinato S, Stacchi C, Lombardi T, Bernardello F, Messina M, Zaffe D.  Biological width establishment around dental implants is  influenced by abutment height irrespective of vertical mucosal thickness: A cluster randomized controlled trial.

Clin Oral Implants Res  2019; 30(7):649-659.

2-Sergio Spinato, Claudio Stacchi, Teresa Lombardi, Fabio Bernardello, Marcello Messina, Sergio Dovigo, Davide Zaffe. Influence of abutment height and vertical mucosal thickness on early marginal  bone loss. An 18-month clinical and radiographic prospective evaluation.

Int  J Oral Implantol 2020;13(3):279-290.

Author Response

# The very significant mean MBL (1.25 mm) described in this article should be underlined. In fact, MBL > 0.44mm at 6 months is an indicator of bone loss progression over time (18 months), as demonstrated by Galindo-Moreno

Galindo-Moreno P, León-Cano A, Ortega-Oller I, Monje A, O Valle F, Catena A. Marginal bone loss as success criterion in implant dentistry: beyond 2 mm. Clin Oral Implants Res. 2015; 26(4): e28-e34.

Response:

Thank you very much for your consideration, we have written a paragraph [In red, lines 250-254]

Although the mean MBL found by us (1.25mm) is higher than that reported by Galindo et al in 2015 [29], who published a 0.6mm loss (mean distal and mesial) at 6 months; and 1.11mm at 18 months (mean distal and mesial). Our study has a mean follow-up of 44.7 months (± 31.4), and if we analyze other studies, such as the one by Palacios-Garzón et al [30], they accept a loss of less than 2 mm in the first year.

And we added references:

  1. Galindo-Moreno P, León-Cano A, Ortega-Oller I, Monje A, O Valle F, Catena A. Marginal bone loss as success criterion in implant dentistry: beyond 2 mm. Clin Oral Implants Res 2015, 26, e28-e34. DOI: 10.1111/clr.12324
  2. Palacios-Garzón N, Mauri-Obradors E, Roselló-LLabrés X, Estrugo-Devesa A, Jané-Salas E, López-López J. Comparison of Marginal Bone Loss Between Implants with Internal and External Connections: A Systematic Review. Int J Oral Maxillofac Implants 2018, 33,580-589: DOI10.11607/jomi.6190.

# In addition, implants with MBL≥0.5 mm during the first year of function showed a 5.43 times higher odds for future peri-implantitis development and implants in patients with smoking habits and a history of periodontitis are at the highest risk of developing peri-implantitis and experiencing implant loss if early bone loss exceeds the threshold of 0.5 mm at 1 year.

Windael S, Collaert B, De Buyser S, De Bruyn H, Vervaeke S. Early peri-implant bone loss as a predictor for peri-implantitis: A 10-year prospective cohort study. Clin Implant Dent Relat Res.

Response:

Once again, thanks for your consideration, we have taken this aspect into account as you have kindly reminded us and Spinato et al. [31] [In red in line 254-256]

However, we must not forget that losses greater than 0.5mm in the first year increase the risk of presenting peri-implantitis in the long term by 5.43 times [31].

We added the reference:

  1. Windael S, Collaert B, De Buyser S, De Bruyn H, Vervaeke S. Early peri-implant bone loss as a predictor for peri-implantitis: A 10-year prospective cohort study. Clin Implant Dent Relat Res 2021, 23,298-308. DOI: 10.1111/cid.13000.

# The role of the Locator abutment height on MBL should be discussed and (eventually) investigated, based on these recent articles: 1- Spinato S, Stacchi C, Lombardi T, Bernardello F, Messina M, Zaffe D.  Biological width establishment around dental implants is  influenced by abutment height irrespective of vertical mucosal thickness: A cluster randomized controlled trial.

Clin Oral Implants Res  2019; 30(7):649-659.

2-Sergio Spinato, Claudio Stacchi, Teresa Lombardi, Fabio Bernardello, Marcello Messina, Sergio Dovigo, Davide Zaffe. Influence of abutment height and vertical mucosal thickness on early marginal  bone loss. An 18-month clinical and radiographic prospective evaluation.

Int  J Oral Implantol 2020;13(3):279-290.

Response:

Our sample is small, only 2 patients have shown peri-implantitis in the follow-up time, in both patients the height of the locator was 2mm. Although different authors relate bone loss to the lower height of the locators, we have not been able to establish any type of relationship. We have added, in red [lines 256-259]

Both in the cases of peri-implantitis, and with regard to the MBL data, we have not been able to relate them to the height of the locator-attachment system used, all of which is that the literature refers to a higher bone order in the lower systems [32,33].

We added the references:

  1. Spinato S, Stacchi C, Lombardi T, Bernardello F, Messina M, Zaffe D. Biological width establishment around dental implants is influenced by abutment height irrespective of vertical mucosal thickness: A cluster randomized controlled trial. Clin Oral Implants Res 2019,30,649-659. DOI: 10.1111/clr.13450
  2. Spinato S, Stacchi C, Lombardi T, Bernardello F, Messina M, Dovigo S, Zaffe D. Influence of abutment height and vertical mucosal thickness on early marginal bone loss around implants: A randomised clinical trial with an 18-month post- loading clinical and radiographic evaluation. Int J Oral Implantol (Berl) 2020,13,279-290. PMID: 32879932
